# Nanocrystalline Cellulose to Reduce Superplasticizer Demand in 3D Printing of Cementitious Materials

**DOI:** 10.3390/ma17174247

**Published:** 2024-08-28

**Authors:** Rocío Jarabo, Elena Fuente, José Luis García Calvo, Pedro Carballosa, Carlos Negro

**Affiliations:** 1Instituto de Ciencias de la Construcción “Eduardo Torrjoja” (CSIC), C/Serrano Galvache 4, 28033 Madrid, Spain; jolgac@ietcc.csic.es (J.L.G.C.); carballosa@ietcc.csic.es (P.C.); 2Chemical Engineering and Materials Department, University Complutense of Madrid, Avda. Complutense s/n, 28040 Madrid, Spain; cnegro@quim.ucm.es

**Keywords:** cement mortar, nanocrystalline cellulose, 3D printing, compressive strength, mechanical resistance

## Abstract

One challenge for 3D printing is that the mortar must flow easily through the printer nozzle, and after printing, it must develop compressive strength fast and high enough to support the layers on it. This requires an exact and difficult control of the superplasticizer (SP) dosing. Nanocrystalline cellulose (CNC) has gained significant interest as a rheological modifier of mortar by interacting with the various cement components. This research studied the potential of nanocrystalline cellulose (CNC) as a mortar aid for 3D printing and its interactions with SPs. Interactions of a CNC and SP with cement suspensions were investigated by means of monitoring the effect on cement dispersion (by monitoring the particle chord length distributions in real time) and their impact on mortar mechanical properties. Although cement dispersion was increased by both CNC and SP, only CNC prevented cement agglomeration when shearing was reduced. Furthermore, combining SP and CNC led to faster development of compressive strength and increased compressive strength up to 30% compared to mortar that had undergone a one-day curing process.

## 1. Introduction

In three-dimensional (3D) printing, a printer nozzle discharges a hardening suspension in successive layers in accordance with a 3D computer model. This is widely used nowadays to produce complex shape geometries with time and cost savings. Cost and sustainability concerns are the main driving forces for the adoption of this technology in the building industry. Even the building of conventional designs, which is hindered in many economies by a lack of skilled labor, now has the potential for an automated solution. Implementation of more tailored, physically efficient designs now has an easier route [1]. Although 3D printing has been established and is effective with polymers, it may also be used with cement materials. 

Material challenges are significant: understanding and controlling early age hydration and its relationship to rheology, incorporating reinforcement, and, overall, the link between processing, material, and performance from both a structural and a durability standpoint are among the major difficulties [2]. There are some limitations in mortar 3D printing: mortar must flow easily and it must have a high workability, but it must have high enough cohesion. Furthermore, hardening kinetics must be fast enough to allow layers to be self-supporting but slow enough to maintain workability and viscosity in values able to be pumped and extruded through the printer nozzle [3]. If the stiffness is high and the cohesion is insufficient, the layers will not be self-supporting or they can have discontinuity flaws [4].

Superplasticizers (SPs) are added to increase the workability of mortar and to control mortar hardening. However, it is difficult to control the effect of an SP on mortar rheology and hydration because it depends on the temperature, composition, and morphology (aggregate size and amount) of the cement [5,6]. Therefore, alternative aids are being studied to improve mortar properties. In addition, SPs are chemical compounds obtained from petroleum-derived raw materials, which limits the sustainability of 3D mortar printing.

Cellulose nanocrystals (CNCs) are produced mainly by acid hydrolysis of the amorphous part of cellulose. Due to their large specific surface area and reactivity, CNCs have gained significant interest because they can alter the mechanical and rheological characteristics of mortar by interacting with the various cement components. Nanocelluloses are still in precommercial development in many of their applications, including the one studied in this research. The data on its costs in the near future, although still uncertain, are expected to be below 3 €/kg [7,8,9,10,11].

CNCs have been studied as a cement rheology modifier [12,13]. According to Cao et al., 2016 [12], CNCs can be employed to improve the flowability of the paste at low doses (0.2% by volume of cement), but at larger dosages (>0.5% by volume of cement), CNCs decrease the flowability of cement paste. The effect on hardening was also studied by Zheng et al., 2023 [14], who proved that CNCs decrease the hydration exothermicity and provide nucleation sites to promote the formation of C-S-H gel.

Most of the studies involving CNCs and cement are aimed at evaluating the reinforcing ability of CNCs in cement material, which is due to the ability of CNCs to bind microcracks and reduce autogenous shrinkage [7,9,15,16]. Nasir et al., 2022 [16] studied the effect of CNCs from deinked office paper waste on cement mortar. Their results show that CNCs can improve compressive and flexural strengths up to 22% and 28%, respectively, but the flowing decreases with CNC dose. This is because of water demand increase due to the high and hygroscopic specific surface of CNCs. Therefore, this effect could be reduced by increasing the water amount in mortar formulation. Guo et al., 2023 [17] studied the effect of CNC doses and CNC sonication on reinforcing efficiency and proved that flexural strength increased up to 10% with a CNC dose around 0.01% when the CNCs were sonicated for 12 min.

Several studies have been carried out to develop mortar 3D printing [18,19,20,21] and to develop cellulosic fiber–cement 3D printing [4], but there is a lack of studies on CNC use in mortar 3D printing. The most significant and recent research in this field has been carried out by Ghantous et al., 2022 [21], who aimed to study the effect of drying in 3D printed cement hydration and how CNCs could contribute to reducing this effect. However, this research does not include the effect of CNCs on mechanical properties of mortar, and it was carried out with cement paste without other components, such as sand, which are used with mortar in 3D printing. Furthermore, there are not published studies on the effect of CNCs on the final properties of 3D printer mortar. Therefore, the aim of this paper is to contribute to filling the knowledge gap on the effect of CNCs on mortar behavior during 3D printing and on the final properties of 3D printed mortar.

## 2. Materials and Methods

To avoid possible influences of supplementary cementitious materials in the performance of the CNCs, type I Portland cement supplied by CEMEX was used (CEM I 52.5R, Nuevo Leon, Mexico). 

An SP additive supplied by BASF, also known as a high-range water reducer, based on the innovative technology of PAE (poly-aryl phosphonic ether) polymers, was used. The main contribution of this additive is the improvement of the rheological behavior of the concrete thanks to the reduction of the plastic viscosity. This SP is used when a well-dispersed suspension of particles is required to improve the flow characteristics (rheology) of suspensions and improve pumpability in applications such as 3D printing. 

Cellulose nanocrystal (CNC) was obtained from a cotton suspension by acid hydrolysis (with H_2_SO_4_) of the cotton followed by dialysis purification to neutral pH, according to the method described by Campano et al. [22]. The yield of the process was 61.1 ± 4.3%. The obtained CNC suspension, with a consistency of 0.72%, was sonicated for 3 min until the turbidity was reduced to the lowest possible value (100 ± 15 NTU) just before being used or freeze-dried. Then, they were characterized by measuring polymerization degree and zeta potential and by TEM (JEOL, Tokyo, Japan), images. Part of the sonicated CNC suspension was frozen at −80 °C and freeze-dried in a LyoQuest_85 (Telstar S.L., Barcelona, Spain) by applying high vacuum (pressure in the chamber was 0.001 mbar) and soft temperature (30 °C) in the trays containing the CNC frozen suspension. In this way, the CNCs were dried without agglomeration. Dry CNCs were used to determine crystallinity.

A siliceous sand (0–2 mm) according to EN 196-1 was also used for the fabrication of mortar samples.

### 2.1. CNC Characterization

The concentration of the obtained CNC suspension was determined by gravimetry at 105 °C. CNC production yield was determined from the dry CNC mass obtained after the dialysis stage and the initial mass of cotton used [22]. During hydrolysis, part of the amorphous cellulose was completely degraded to soluble compounds. The amount of organic soluble compounds was determined from the chemical oxygen demand (*COD*) of the supernatant just after the hydrolysis stage and by means of Equation (1), developed by Wang et al., 2012, for cellulose [23]. Then, the total mass of *DAC* was calculated by multiplying by the total volume of reaction, and it was expressed as a percentage on the total cotton used.
(1)$$DAC\;(\frac{mg}{L})=\frac{COD\;(\frac{mg}{L})}{1185}$$

Zeta potential of the CNC suspension diluted to 0.005 g/L and at pH 6 was determined by means of a 90PlusZeta device (Brookhaven Instruments Corporation, Holtsville, NY, USA). Polymerization degree was estimated from the intrinsic viscosity measured by following the ISO 5351:2010 standard method [24], which is based on forming a complex of cellulose with cupriethylenediamine and measuring the intrinsic viscosity (*µ*) of that complex with a glass tube viscometer. Polymerization degree (*PD*) was estimated from Equations (2) and (3), proposed by Henriksson et al., 2008 [25]. Constant values (0.42 and 2.28) are related to the difficulty of dispersing molecular dissolutions, which increases with *PD*.
(2)$$PD=\frac{\mu }{0.42}\;if\;PD<950$$
(3)$$PD={\left(\frac{\mu }{2.28}\right)}^{\frac{1}{0.76}}if\;PD>950$$

AN X-ray diffractogram (XRD) of the dry CNCs was obtained with a Philips X’Pert MPD X-ray diffractometer with an autodivergent slit fitted with a graphite monochromator using Cu-Kα radiation operated at 45 kV and 40 mA. The XRD patterns were recorded at a scanning speed of 0.6/min. The crystallinity index (CrI) was estimated by following Segal’s method [26] (Equation (4)) from the intensity at the peak at a 2θ of 22.5°, *I*_200_, and the intensity at a 2θ of 18°, *I_am_*.
(4)$$CrI(\%)=100\frac{I_{200}-I_{am}}{I_{200}}$$

Images of CNCs were taken by transmission electronic microscopy (TEM) at the Spanish National Centre of Electronic Microscopy. A total of 15 images were taken of different zones of the sample to select the most representative for the CNC sample. The microscope used was a JEOL JEM 1400 plus TEM (JEOL, Tokyo, Japan), operated at 100 kV accelerating voltage, with a CCD camera Orius SC200 (Gatan, CA, USA), which has a 2048 × 2048 pixel resolution and a pixel size of 7.4 µm. To take the images, a small portion of CNC suspension was diluted to 5 ppm. Then, the CNCs were fixed to a 200-mesh copper grid covered with a continuous layer of 10 nm Formvar and stabilized with 1 nm evaporated carbon film. The CNCs were fixed to the grid by means of Poly-L-Lysine (0.02 mL of Poly-L-Lysine 10% solution was added to the grid and dried at 60 °C before adding 0.01 mL of diluted CNC suspension) to avoid CNC agglomeration. Size distribution was obtained by image analysis with ImageJ software (version 1.49q).

### 2.2. Flocculation Trials

A focused beam reflectance measurement (FBRM) G400L probe supplied by Mettler Toledo, Columbus, OH, USA, was used to monitor the flocculation process and to determine the floc properties. The FBRM monitors the chord length distribution of the particles in suspensions in situ and in real time. From each chord length distribution, the system can calculate different statistics, such as the mean of the distribution (mean chord length) and the total number of particles detected per second (total number of counts per second). The principle of the measurement and the details of the applied methodology were described by the authors in previous research [27].

In a typical trial of flocculation, the probe was immersed into 400 mL of distilled water and stirred at 800 rpm. Cement was added until reaching a concentration of 20 wt.%. After assuring cement dispersion, the stirring was varied cyclically between 400 rpm and 800 rpm. Then 10 min after cement addition, stirring intensity was reduced to 400 rpm (reflocculation). The intensity was increased to 800 rpm to break down the formed flocs (deflocculation) 5 min later. Then 20 min after cement addition, CNC or SP dose was added while stirring was 800 rpm, and one minute later the SP was added, if required, in the cases in which CNC and SP were used together. Table 1 shows the different tried doses of CNC and SP. Then 5 min after that, stirring was reduced again to 400 rpm to induce the reflocculation of the system. In all cases, 80 g of cement and 400 g of water were used per trial. Every 5 min, the intensity was varied cyclically: 800-400-800-400 rpm [28]. Trials were carried out in duplicate. 

In this test, the direct effect of the addition of CNC or SP to the cement suspension at 800 rpm and its effect on reflocculation can be observed and compared with the first flocculation cycle without additive.

### 2.3. Tests in Mortar Specimens

The effects of CNCs in suspension and CNCs lyophilized, and their interaction with SPs, on mechanical properties of cementitious materials were studied in modified standard mortars. Table 2 shows the mix compositions of the fabricated mortars. Mortars were prepared by following the standard procedure defined in EN-196-1 considering the next modifications: CNCs in suspension (and SPs, if any) were added directly with the water while lyophilized CNCs were previously mixed with the cement. The content of lyophilized CNCs was double that of suspended CNCs since higher effectivity of suspended CNCs was detected in previous studies.

The addition of an SP decreased water demand of the mortar. This decreases the yield stress, increasing flowability. To keep a constant flowability, water amount was adapted to cover the decrease in water demand due to the absence of an SP since good flowability is required for 3D printing but reducing the water/cement ratio improves mechanical properties. The flowability of fresh mortar mixes was controlled through the flow table test according to the standard UNE-EN 1015-3 [29]. This consists of measuring the horizontal spread that a cement mortar experiences because of successive dynamic impacts. The amount of water added was adjusted to obtain a flow value between 180 and 200 mm, which is acceptable for 3D printing. Nine prismatic mortar samples of 40 mm × 40 mm × 160 mm were casted per mix to evaluate the mechanical performance evolution after 1, 28, and 90 days of standard curing (relative humidity of 98% and temperature of 20 ± 2 °C). The flexural and compressive strengths of the hardened mortars were determined according to the EN 196-1 standard [30]. On every test day, the prismatic mortar samples were submitted first to a three-point bending flexural strength test. The resultant halves of each prismatic mortar sample were then subjected to a compressive strength test. Therefore, the outcome is three flexural and six compressive strength measurements per mix and per age.

Moreover, to evaluate if the incorporation of CNCs in suspension promotes delays in the setting times of cementitious materials, the setting times (both initial and final) of REF and 0.05 S were determined according to EN-196-3. This characteristic was not evaluated in the samples with an SP to avoid any influence of the chemical admixture in the results obtained.

## 3. Results and Discussion

### 3.1. CNC Characterization

Table 3 shows the results of CNC characterization. High CNC production yield and low DAC were obtained because of the high crystallinity and purity of cotton cellulose. Crystalline cellulose is harder to hydrolyze than the amorphous part, but the fact that the yield was lower than the CrI of the raw material (90%) indicates that part of the crystalline cellulose was degraded too. The CrI was higher than the CrI of the cotton, as expected, which indicates that the presence of amorphous regions on CNC was negligible. The high chemical strength of the crystalline part of cellulose and the high crystallinity of cotton fibers (90%) allow obtaining a high polymerization degree for CNCs. This value was slightly lower than that obtained by Campano et al., 2021 [31] for CNCs obtained from cotton with the same procedure, but the yield and the CrI were slightly higher and PD slightly lower. This indicates that a higher selectivity was achieved in this hydrolysis and a higher amount of the amorphous part was removed with lower damage to the crystalline part. Chen et al., 2015, reported a high sensitivity of hydrolysis yield and DAC with acid concentration near the optimal (60%). Therefore, slight variations between batches are assumed as indicated by the standard deviation of the yield in Table 3, reaching up to 10%. The negative zeta potential is mainly due to the sulphate ester groups at the CNC surface [32].

The CNCs were well-dispersed rod-like rigid particles (Figure 1). Few small aggregates were formed. The use of L-lysine in sample preparation for TEM imaging allowed us to keep the aggregation degree without variation, representing the aggregation degree of CNCs used in the trials. The length distribution of CNCs is quite wide (Figure 2), with 90% of CNCs shorter than 280 nm and thinner than 27 nm.

### 3.2. Effect of CNCs on Cement Suspension Behavior

When a cement suspension is stirred and stirring intensity is decreased, the cement particles aggregate in a natural way without any flocculant addition [33]. Figure 3 shows that the MCL (mean chord length) of cement suspension increased around 1 µm when stirring intensity decreased from 800 rpm to 400 rpm. The initial MCL value was recovered by increasing again the stirring speed. This behavior is fully reproducible as shown by the evolution of the MCL during the three stirring cycles shown in Figure 3. A small variation can be observed in the first cycle due to the small differences in the first part of the cement dispersion process, but after that first cycle, all the curves reached a very similar mean chord length (Figure 4). Therefore, CNCs were added in that second cycle. When CNCs were added (Figure 4, time 15 min), the value of MCL slightly decreased and the aggregation of cement was partially impaired, as shown by the lower MCL values obtained at 400 rpm. A small dose of CNCs (0.025%) was high enough to reduce cement aggregation by half when the stirring intensity varied from 800 rpm to 400 rpm. The addition of CNC doses greater than 0.025% decreased the MCL at the time of addition. This was not due to dilution, since the CNC suspension volumes were small (i.e., for 0.025%, 2.7 mL of CNC suspension was required, as the volume of cement paste was 400 mL). The CNC particles were too small to be detected by the FBRM as they were fully dispersed. Then, the decrease in MCL value when CNCs were added indicates the increase in the dispersion of the cement suspension. This agrees with the observations carried out by Cao et al. [12]. CNCs adsorb on cement particles and cause steric and electrostatic repulsion among cement particles, increasing dispersion. This is due to the high anionic charge of CNCs as their zeta potential is around −27 mV. This completes the observations made by Montes et al., 2020 [13], who observed a decrease in yield stress of cement paste by low doses of CNC that was influenced by both zeta potential and length. They explained it as a result of steric or electrosteric stabilization, but they did not have experimental evidence for that. Results shown in Figure 4 are evidence of the theoretical dispersion ability of CNCs in cement suspensions. Furthermore, Montes et al., 2020 [13] observed that the yield stress increased by high doses of CNC (higher than 0.6%), which shows that there are other mechanisms for yield stress modification. In this case, a water demand increase due to the high hygroscopicity of CNCs could be the explanation for the effect in yield stress, as it has been observed that CNCs retain water molecules around their surface due to the large and hydrophilic surface [14,15,16].

The dispersing effect increased with the dose of CNCs and, when the dose of CNCs was over 0.1%w, cement aggregation was fully avoided. Therefore, CNCs acted as a disperser aid in cement suspension. Figure 5 relates the increase in the MCL when the stirring intensity decreased to 400 rpm after the addition of CNCs (second cycle). This MCL increment is defined as the difference between the mean of the MCL values during the 3 min before the CNC addition and the mean of values of MCL during 3 min after reducing stirring intensity to 400 rpm. The dispersing effect is higher the lower that difference is.

Figure 5 shows that there is a clear relation among CNC dose and MCL increase. The dispersing effect increased with the CNC dose up to a dose of 0.1% of CNCs. Higher doses of CNCs (i.e., 0.25%) did not increase dispersion.

In view of the results in Figure 3 and Figure 4, the selected doses of CNCs to continue with the tests were 0.05% and 0.1% CNCs because the dispersion effect is like that caused by 0.25% CNCs, but the cost is lower than half of that for adding 0.25% CNCs. Furthermore, the MCL with 0.1% CNCs remained stable when stirring decreased to 400 rpm. There is not a clear correlation between the mean chord length and strength, but the higher dispersion of cement contributes to a more homogeneous matrix, which favors the mechanical properties’ development. However, an excess of CNCs increases the costs and the water demand of cement mortar. 

### 3.3. Effect of SPs on Cement Suspension

Figure 6 shows the evolution of the MCL obtained by adding an SP. As shown by the increase in the number of counts and the decrease in the MCL, the addition of the SP caused a dispersion of the cement suspension. This was expected and shows that the SP was efficient. There are various studies on the effect of SPs in cement and on how they interact with suspended particles and with their hydration products [34,35,36,37]. In general, there is consensus that an SP causes an increase in the dispersion of cement particles, making the paste flow better [34].

However, when stirring speed decreased from 800 rpm to 400 rpm, the MCL decreased and the total number of counts increased, and this was reversible: they recovered their values by increasing stirring speed to 800 rpm. This was the opposite behavior than that observed for cement suspension without SP (Figure 3). This increase in the total number of counts and the decrease in MCL at 400 rpm could indicate a dispersion, but it is controversial because the shearing forces are higher at 800 rpm than at 400 rpm. Therefore, a deep analysis is required to explain this phenomenon.

Considering that the SP dispersed the suspension, which is demonstrated by the evolution of the MCL and the number of counts after the SP addition (Figure 6), the reversal behavior of MCL and total number of counts could be due to the interaction of the SP preferentially on the small particle aggregates inside the measurement interval; however, the chord length distributions (Figure 6b) show that the number of counts in the chord length interval from 20 to 100 µm slightly decreased when stirring intensity decreased to 400 rpm, and there is a real displacement of the distribution towards lower chord lengths, with a soft increase in its area. Furthermore, the effect of the SP at 400 rpm was higher than the effect caused by the SP addition at 800 rpm. This could indicate that the adsorption of the SP on the particles’ surface is reversible and adsorption forces are weak. Then, at 800 rpm, part of the SP was not adsorbed on cement particles because of the high shearing forces promoting desorption. However, at 400 rpm, a higher adsorption of SP increased cement dispersion. The repeated cycles showed that the adsorption and dispersion were reversible at least during the first 50 min. This has been observed for some SPs, as for example, polycarboxylate dispersants [38], and was deeply studied by Zhang et al. [39]. Most of the SP was absorbed by means of electrostatic interaction with Ca^2+^ ions released from the cement particles during early hydration [40]. Since the cement suspension mainly consists of four different minerals (tricalcium silicate, C3S, dicalcium silicate, C2S, tricalcium aluminate, C3A, and tetra calcium aluminum ferrite, C4AF) the adsorption of the SP on all the minerals can be different as there is a higher affinity of the SP for C3A and C4AF particles than for others [41].

A lower SP dose (0.5%) was tried, and the evolution of MCL was very similar (Figure 7). Therefore, there was not a significant improvement by using a dose of SP of 0.85% instead of 0.5%. A dose of SP of 0.5% over cement was enough to induce cement dispersion.

The effect on cement hydration and mechanical properties depends on the SP itself and the dose used. There are studies that show that SP interferes with the hydration process, in some cases slowing it down, while in other cases reducing the crystallinity of the hydration products [40,42]. However, the use of SP achieves improvements in mechanical properties using a correct dose of SP, not because of the SP itself but because of the lower demand for water and better workability of the cement containing SP.

Figure 8 indicates that the dispersing effect of the CNCs was lower than that of the SP, although the dose was lower too. Doses of CNCs were limited because of their cost and their increase in water demand; thus, the maximum dose tried was 0.25%. Furthermore, the effect of using 0.25% and 0.1% on the dispersing effect was the same (Figure 5). Therefore, increasing the CNC dose was not worthwhile. On the other hand, the dispersing effect was exerted by adding the CNCs but did not vary by reducing the agitation. This indicates that particles were likely stabilized by increasing anionic charge in particles’ surface since CNCs were anionic, especially at high pH. The SP, however, allowed some aggregation of particles by varying shear forces, i.e., during printing.

### 3.4. Synergic Effect of SP and CNCs on Cement Suspension

Figure 9 shows the effect of the combined addition of CNCs and the SP on the cement suspension. Two doses of CNCs were combined with a dose of 0.85% SP: 0.05% and 0.1%. In both cases, the effect of the intensity of agitation on the evolution of the TMC was almost completely reduced, registering a stable value of the MCL despite the decrease or increase in the stirring speed, which indicates that the presence of CNC stabilized the particles of the suspension, reducing its aggregation when the stirring intensity decreased.

It is noteworthy that the effect of the lowest dose of CNCs, 0.05%, was enhanced by the presence of SP. The addition of CNCs to 0.05% itself did not prevent the aggregation induced by the decrease in shearing forces; the MCL still increased when stirring intensity decreased to 400 rpm (Figure 4). However, in the presence of the SP, the cement suspension behaved similarly to that containing 0.1% dose of CNCs.

The adsorption of CNCs on cement was stronger than that of the SP, as shown by the stable MCL at varying shearing (Figure 8). The higher size of CNC compared to SP molecules promotes cement suspension stabilization by steric repulsion. A dose of 0.05% of CNCs is not high enough to keep the suspension dispersed by steric repulsion, but the SP addition increased cement particles dispersion by means of electrostatic repulsion. It is known that polycarboxylic acids can form esters with cellulose [43]. This reaction attached the SP molecules to the adsorbed CNCs, increasing steric and electrostatic repulsive forces, which kept the cement suspension dispersed under different shearing forces. A dose of 0.05% of CNCs fixed an amount of SP molecules high enough to avoid particle aggregation, and an increase in CNC dose did not have any differential effect. 

### 3.5. Effect of CNCs on the Performance of Cementitious Materials

The effect of the addition of CNCs in two different ways (suspended or lyophilized) on the fresh state and the mechanical strength evolution of cementitious materials was evaluated in the fabricated mortars. Table 4 shows the flow (consistency) of the fresh mortars measured by means of the flow table test. All the mortar compositions were adjusted to obtain a similar consistency, between 180 and 200 mm, which is an acceptable range to be 3D printed. In the samples without SP, the addition of CNCs in both ways increased the flow, which agrees with the dispersion effect promoted by CNCs and verified in Figure 4 and Figure 5. Although the consistency obtained in all the mortars was very similar, those of the mortars with SP were slightly lower due to their lower w/c ratio, which was 0.4 to avoid possible segregation problems. The lyophilized CNC addition in mortars with SP increased flow too, which agrees with the dispersion effect obtained when combining an SP and CNCs (Figure 9). The effect in the case of the CNC suspension was negligible in the presence of the SP. Therefore, the lyophilized CNC could promote slightly higher fluidity in the mortar samples, although the difference was quite low (5 mm). Regardless, this higher increase in the flow when using lyophilized CNCs is due to the higher CNC dose compared to suspended CNCs (0.1% vs. 0.05%). 

Table 5 shows the initial and final setting times measured for the reference mortar (REF) and that with CNC (0.05S) samples. The two types of mortar did not significantly differ in their setting times except for the mortar with CNCs in suspension having a slightly quicker initial setting time. In this sense, although the incorporation of CNCs in suspension promoted steric repulsion between the cement particles (Figure 4 and Figure 5), it did not delay the setting times. On the contrary, the slightly faster initial setting time could improve the mortar performance for 3D printing use. 

Figure 10 shows the flexural strength evolution of the fabricated mortars and Figure 11 the compressive strength. As expected, the mortars with SP had higher mechanical strength values than the same ones without SP due to the lower w/c ratio of the latter. Although the flexural strength is not the main mechanical property of cementitious materials for 3D printing, the reference mortar with SP seemed to show the highest values during all the test periods (Figure 10). Previous studies have shown an increase in the flexural strength when adding CNCs in contents lower than 0.2 vol.% [12]. This increase was attributed to an increase in the degree of hydration due to two phenomena: (1) steric stabilization was responsible for dispersing the cement particles [13,15]; (2) CNCs provided a channel for water transporting through the hydration products ring to the unhydrated cement particle and thereby improved hydration [15,16]. The dispersion effect was demonstrated in this research too, but the second phenomenon should be evaluated, although it would be expected to occur and agrees with Nasir et al. and El Bakkari et al. [16,44]. Thus, since the flexural strength did not increase in the present study, the fabrication process of the CNCs could play a relevant role. In this sense, certain parameters such as the length and the aspect ratio of the CNCs should influence the flexural strength measured in cementitious materials. The CNC length is too short to allow them binding microcracks, which explains the low effect on flexural strength. In fact, Hisseine et al. [45,46] also demonstrated the increase in the flexural strength of cement pastes by using cellulose filaments. Fresh cement pastes were used for the research carried out by other authors [13,15,43,47], while the present study was made with mortars containing sand. Hence, the addition of the sand is expected to promote certain modifications in the synergistic properties between the hydrated cement matrix and the CNCs. 

While there were no significant differences in compressive strength values of the samples with superplasticizer, the samples with suspended CNCs gained strength more quickly than the reference sample and the samples with lyophilized CNCs, despite the lower CNC content. In fact, the highest efficiency of the addition of suspended CNCs is corroborated in the samples without SP in the short and in the long term. 

At 1 day of curing, the compressive strength of mortar with 0.05% CNCs (0.05S) was 30% higher than that for the reference (REF) mortar and 40% higher than that of the mortar with lyophilized CNCs (0.1L). At 90 days of curing, the corresponding values were 15% and 9% higher. Thus, the inclusion of CNCs in suspension promoted two effects: (i) a faster hydration of the anhydrous cement in the short term, and (ii) a long-term increase in the compressive strength value. In fact, this last effect was also detected in the mortar with lyophilized CNCs since it showed a compressive strength value 6% higher than that for the reference mortar (without CNCs). It is remarkable that the long-term increase in the compressive strength when using CNCs was not detected in the samples with SP since similar values were obtained in the three mortar types containing SP. 

The enhancement of the long-term compressive strength agrees with previous studies made in self-consolidating concretes and in ultra-high-performance concretes by using cellulose filaments [44]. However, the compressive strength of cement pastes with cellulose filaments was adversely affected because of air entrainment and filament agglomeration [45]. In fact, the strength improvement detected in self-compacting concretes (16%) was very similar to the one detected in the mortars fabricated in the present study with suspended CNCs (15%). The effect of cellulose filaments on enhancing the mechanical strength was found to stem from improvements in microstructure properties of the cement paste system, mainly by the promotion of an increase in the cement degree of hydration [46]. This increased degree of hydration was attributed to the hydrophilicity and hygroscopicity of cellulose filaments leading to, respectively, water retention and water release, thereby providing supplementary water (during hydration) for further reaction of anhydrous cement grains. In this sense, this last aspect can explain the increase in the compressive strength detected at 90 days in the mortars with CNCs without SP with respect to the reference mortar without SP (see Figure 11). The fact that the long-term increase in compressive strength was not detected when using CNCs plus SP could be related to some of the interactions observed in the previous section. In any case, recent studies have also detected this enhancement of the long-term compressive strength when using 0.15% of cellulose filaments [48]. Thus, the CNCs used in the present study influenced the compressive strength in a similar way to cellulose filaments.

## 4. Conclusions

The interaction between CNC particles and cement has a dispersing effect on the cement suspension and reduces or avoids its aggregation when hydrodynamic forces decrease. Although the dispersing effect is lower than that observed with an SP, it does not prevent cement hydration and maintains stable dispersion even when the intensity of agitation decreases. The SP does not prevent some reversible aggregation of particles as a consequence of SP adsorption–desorption processes driven by shearing forces.

The use of CNCs could be a sustainable alternative to the use of SP at low doses, although it does not decrease water demand and it is necessary to carry out studies on the physical characteristics of the cement admixed with CNCs. The increase in the flow (flow table test) due to the incorporation of CNCs was corroborated in mortars without SP. Moreover, this increase in the flow during the fresh state did not influence the setting time of the resulting mortars. The replacement of SPs (from fossil raw materials) by CNCs (from renewable cellulosic raw materials) would increase the sustainability of 3D printing of mortar. 

The combination of an SP and CNCs had a synergistic effect so that the dose of CNCs required to achieve dispersed stable cement suspension was reduced when the SP was used, and the development of compressive strength was faster when the SP and CNCs were used together, which can increase the ability of each layer to support the layers printed over it.

The supplementary cementitious materials of other kinds of cement could modify the performance of CNCs. The research described in this paper opens the door to study new synergistic effects with other cements. 

## Figures and Tables

**Figure 1 materials-17-04247-f001:**
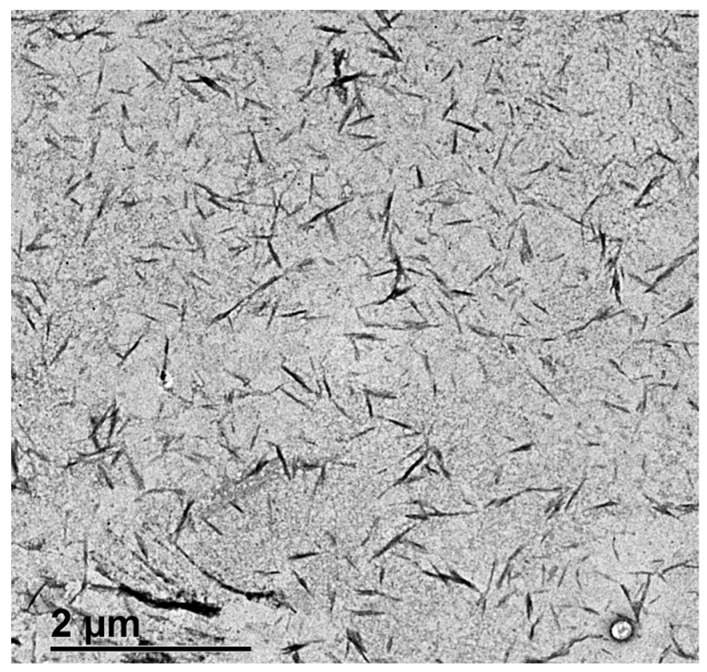
TEM image of CNCs.

**Figure 2 materials-17-04247-f002:**
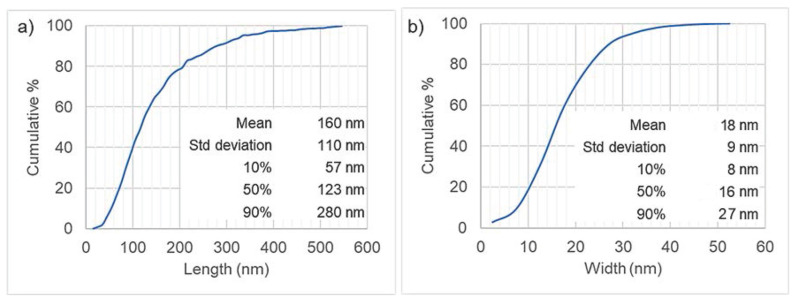
(**a**) Length distribution and (**b**) width distribution, both obtained from the analysis of 15 TEM images.

**Figure 3 materials-17-04247-f003:**
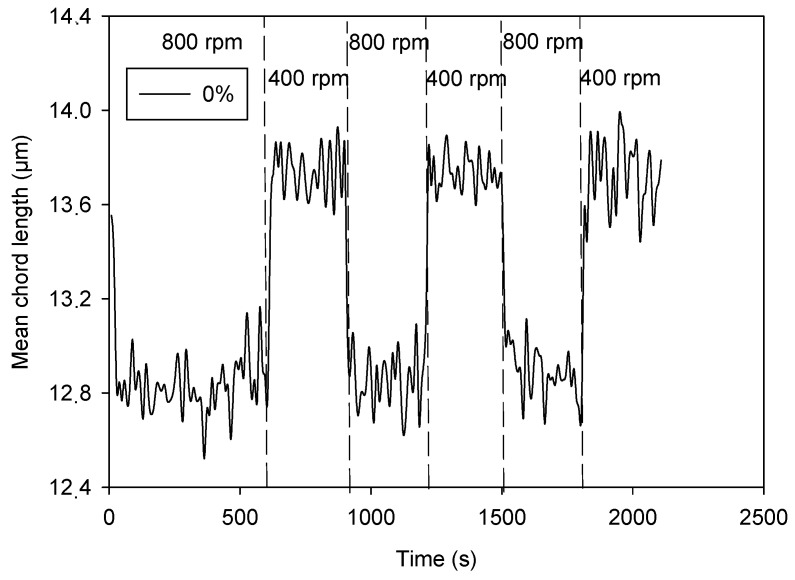
Aggregation and dispersion of cement suspension under stirring.

**Figure 4 materials-17-04247-f004:**
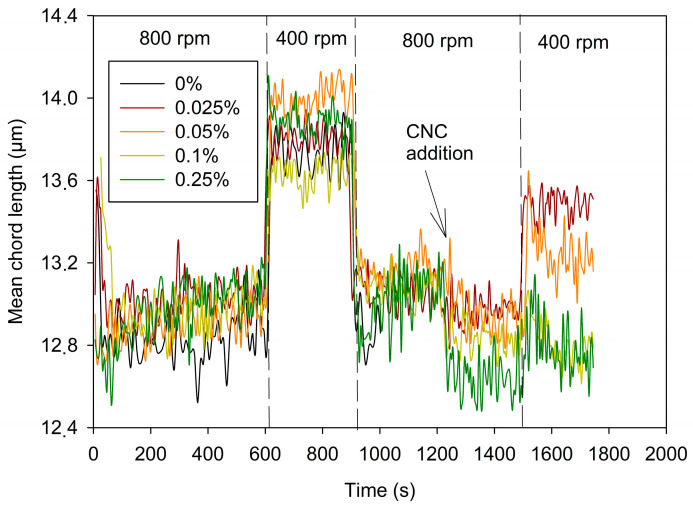
Mean chord length evolution. Effect of CNC dose.

**Figure 5 materials-17-04247-f005:**
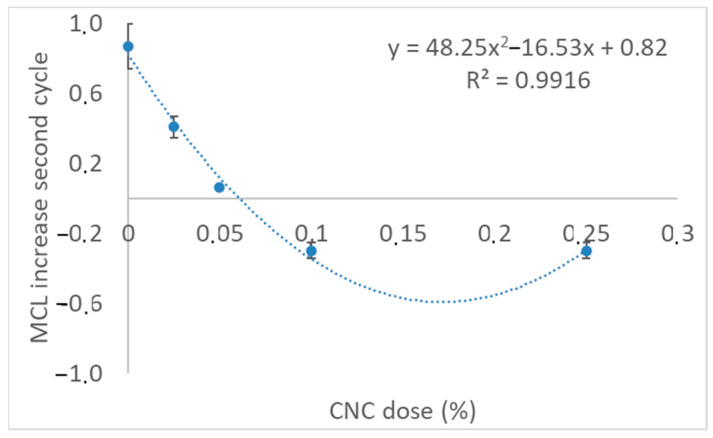
Effect of the CNC dose on cement aggregation.

**Figure 6 materials-17-04247-f006:**
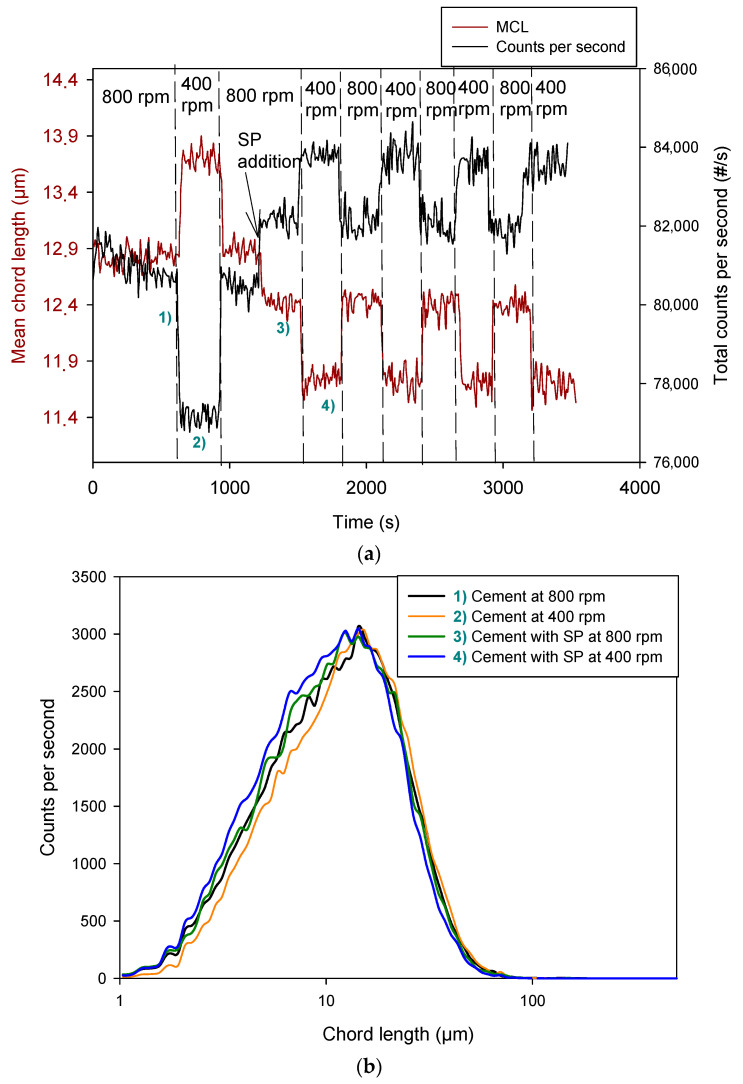
(**a**) Evolution of MCL and total number of counts in presence of 0.85% SP; (**b**) chord length distributions in different moments of the trial.

**Figure 7 materials-17-04247-f007:**
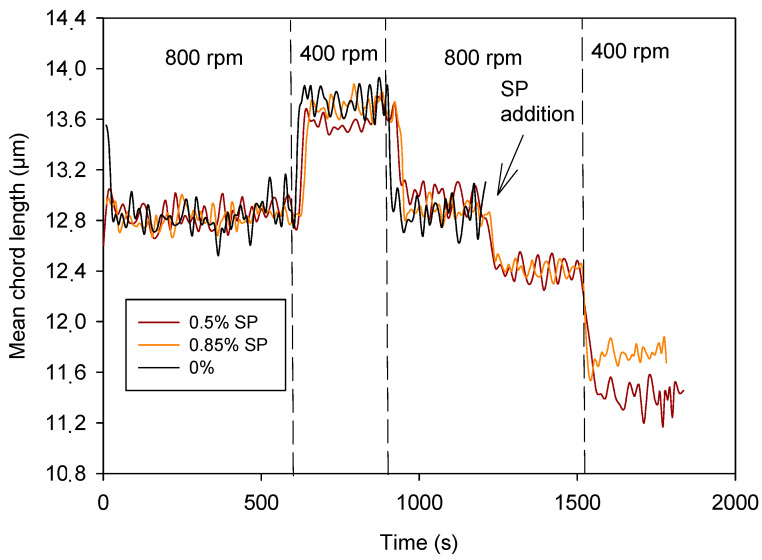
Effect of SP dose on cement dispersion.

**Figure 8 materials-17-04247-f008:**
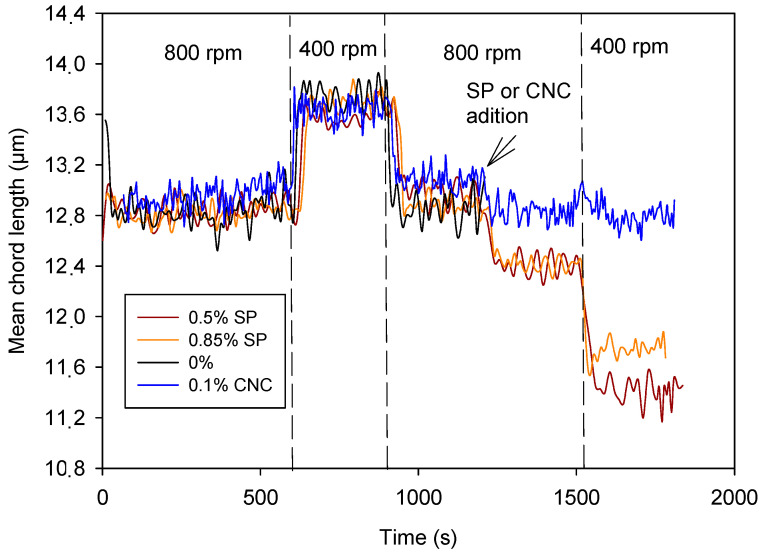
Comparison between the effect of the SP and CNCs on cement.

**Figure 9 materials-17-04247-f009:**
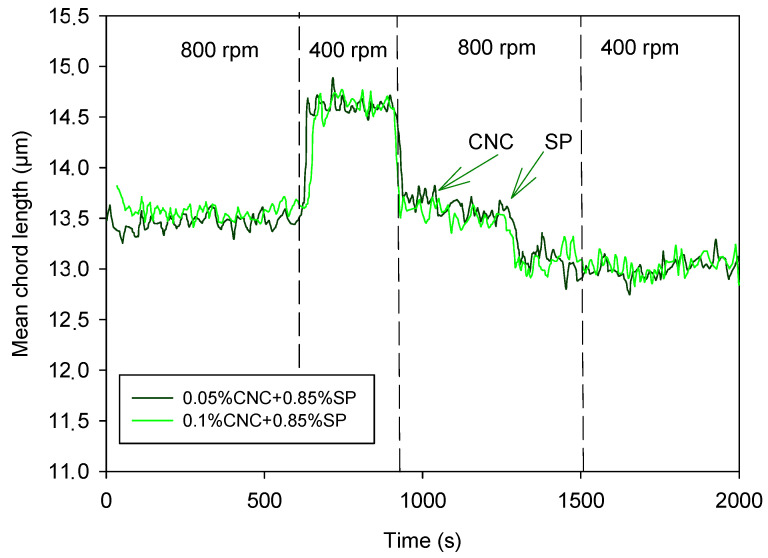
Synergic effect of CNCs and SP.

**Figure 10 materials-17-04247-f010:**
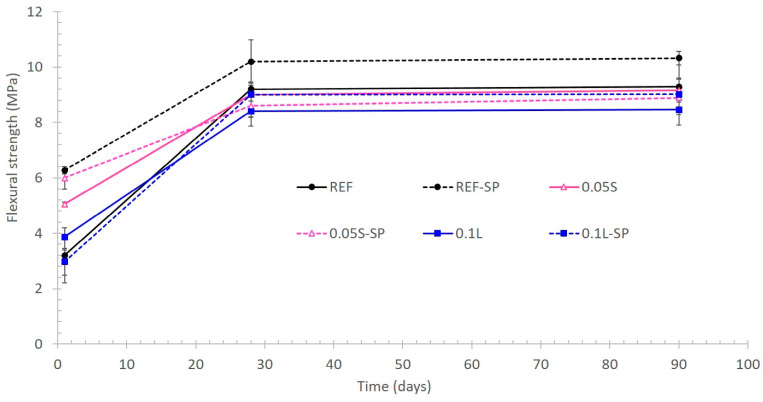
Flexural strength evolution of the fabricated mortars.

**Figure 11 materials-17-04247-f011:**
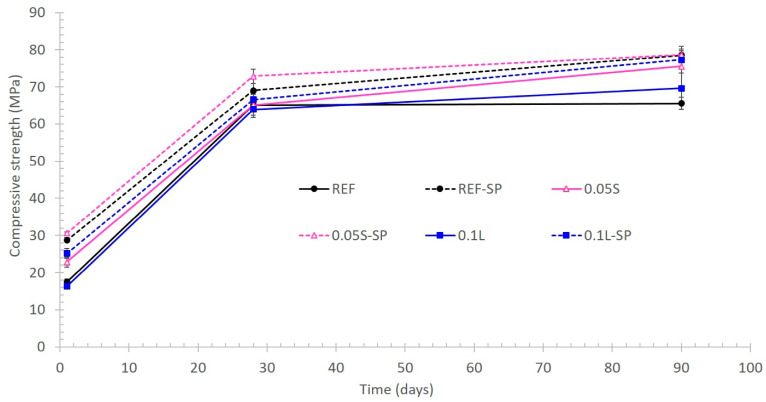
Compressive strength evolution of the fabricated mortars.

**Table 1 materials-17-04247-t001:** Amounts and doses of each material used in flocculation trials.

Trial Name	SP (%)	CNC (%)	SP ^1^ (g)	CNC ^2^ (g)
REF (0%)	0	0	0	0
0.025% CNC	0	0.025	0	0.02
0.05% CNC	0	0.05	0	0.04
0.10% CNC	0	0.1	0	0.08
0.25% CNC	0	0.25	0	0.2
0.5% SP	0.5	0	0.4	0
0.85% SP	0.85	0	0.68	0
0.25% CNC + 0.5% SP	0.5	0.25	0.4	0.2

^1^ Grams of commercial SP product; ^2^ grams of dry CNCs. Concentration of CNC suspension was 7.2 g/L.

**Table 2 materials-17-04247-t002:** Composition of the fabricated mortars.

Raw Material	REF	REF-SP	0.05S	0.05S-SP	0.1L	0.1L-SP
Cement (g)	450	450	450	450	450	450
Water (g)	225	180	225	180	225	180
Siliceous sand (g)	1350	1350	1350	1350	1350	1350
SP (g)	---	3.8	---	3.8	---	3.8
Suspended CNCs (dry g)	---	---	0.41	0.41	---	---
Lyophilized CNCs (g)	---	---	---	---	0.82	0.82

**Table 3 materials-17-04247-t003:** Characteristics of CNCs.

Yield (%)	61.1 ± 6.0
DAC (%)	32.6 ± 2.5
PD	224.2 ± 4.1
Zeta potential (mV)	−27 ± 4
CrI (%)	94.4 ± 0.5
Length (nm)	160 ± 110
Width (nm)	18 ± 9

**Table 4 materials-17-04247-t004:** Consistency of the fresh mortars.

Raw Material	REF	REF-SP	0.05S	0.05S-SP	0.1L	0.1L-SP
Consistency (mm)	195	185	200	185	205	190
w/c	0.5	0.4	0.5	0.4	0.5	0.4

**Table 5 materials-17-04247-t005:** Setting times of REF and 0.05S mortars.

Setting Time (s)	REF	0.05S
Initial	160	150
Final	440	440

## Data Availability

The raw data supporting the conclusions of this article will be made available by the authors on request.

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
