# Peer review of "Nanocrystalline Cellulose to Reduce Superplasticizer Demand in 3D Printing of Cementitious Materials"

_materials, 2024, doi:10.3390/ma17174247_

Round 1
Reviewer 1 Report
Comments and Suggestions for Authors
The Manuscript "Nanocrystalline cellulose to reduce superplasticizer demand in 3D printing of cementitious materials" is well written, materials and method are properly described.
This topic is very interesting for engineering practice.
Below are some comments and questions to the authors:
1. CEM I 52.5R was used in the tests. Why was this cement chosen? Will similar results be obtained, e.g. choosing CEM II or CEM V (lower CO2 emissions)?
2. What are the costs of making this mortar? Is CNS cheaper than SP?
3. Line 178: Table 1. What means "*" for SP? Also missing are indexes 1 and 2, which are referred to under table 1.
Author Response
Comment 1. CEM I 52.5R was used in the tests. Why was this cement chosen? Will similar results be obtained, e.g. choosing CEM II or CEM V (lower CO2 emissions)?
Response 1: Thank you for pointing this out. CEM I was used to avoid possible synergies or influences of supplementary cementitious materials in the performance of the CNC. For sure, the use od CEM II or CEM V could modify the perofrmance of the CNC and this aspect must be evaluated in future studies. We have modified the sentence in page 2 line 87 to clarify this for the readers. In addition, we have included this information in lines 543-545 in the conclusions part.
Comment 2: What are the costs of making this mortar? Is CNC cheaper than SP?
Response 2: Thank you for pointing this out. We agree with the reviewer about the concern on the costs. Nanocelluloses are a new family of products still in pre-commercial development in many of their applications, including the one studied in this research. Therefore, the data on their costs in the near future, although still uncertain, are expected to be below 3€/Kg. In this research, a synergistic effect with SP and some improvements in the hardened state have been demonstrated; Therefore, it is possible to improve the process and its environmental impact. This has been explained in the introduction part, page 2, lines 54-57.
Comment 3: Line 178: Table 1. What means "*" for SP? Also missing are indexes 1 and 2, which are referred to under table 1
Response 3: Thank you for pointing this out. We agree with this comment and we have corrected the notes and indexes in table 1 (page 4, line 185).
Reviewer 2 Report
Comments and Suggestions for Authors
The use of additive technologies currently covers almost all areas of human activity. 3D printers and their products can be found in space, medicine and even construction. Various materials are used for printing on such devices. However, they all have one thing in common - rheology. For such materials, it is necessary that the viscous and elastic properties are in a certain range, and the properties of the extruded material must provide rigidity for applying subsequent layers to them. It is not always possible to achieve such properties when using, for example, one polymer. Therefore, various additives are introduced into the matrix, the task of which is to provide the required rheological behavior. Such additives can be, for example, plasticizers or sensitive to UV radiation. This manuscript proposes to use CNC as an additive.
96-97. Perhaps the sequence of studies was different? Usually they begin with studying the degree of polymerization of cellulose.
111. COD needs to be deciphered.
128. "𝑃𝐷0.76" - the reader will not understand what the authors mean. It is also important to explain the use of different denominators 2.28 and 0.42.
238. Based on Table 3, we can talk about smaller values, the so-called 32 nm instead of 35 nm (250 nm instead of 280 nm), otherwise we need to correct the data in the table and Figure 2.
I don't understand why the curves change places in Figure 4 after the first cycle at 400 rpm? The curve for the system with 0.025% CNC is higher compared to 0.05% CNC.
396. We can add CNC to 0.05%.
542. Add a period at the end of the sentence.
The conclusions and the list of references are compiled in accordance with the requirements of the journal.
The manuscript explains in a simple form the method of replacing SP with CNC, i.e. it suggests a transition from the use of synthetic materials to natural and constantly renewable ones. I think this is interesting, especially since the results obtained by the authors have a positive assessment. I think the manuscript deserves the right to be published.
Author Response
Comment 1: 96-97. Perhaps the sequence of studies was different? Usually they begin with studying the degree of polymerization of cellulose.
Response 1. Thank you for pointing this out. We agree with this comment; actually, polymerization degree and zeta potential were determined before taking TEM images. The order have been changed in the sentence (page 3 lines 100-101).
Comment 2: 111. COD needs to be deciphered.
Response 2: Thank you for pointing this out. We agree with this comment. COD has been deciphered in page 3, line 115.
Comment 3: 128. "??0.76" - the reader will not understand what the authors mean. It is also important to explain the use of different denominators 2.28 and 0.42.
Response 3: Thank you for pointing this out. We agree with this comment. Equation 3 have been expressed in a more comprehensive way and the explanation of the denominators have been included (page 3, lines 129-130 and 134).
Comment 4: 238. Based on Table 3, we can talk about smaller values, the so-called 32 nm instead of 35 nm (250 nm instead of 280 nm), otherwise we need to correct the data in the table and Figure 2.
Response 4: Thank you for pointing this out. We agree with this comment. We have corrected the values of the mean and standard deviations in table 2 and in figure 2 according to the distributions shown in figure 2, and the value of 90% percentile of width in the text (page 6 line 244).
Comment 4: I don't understand why the curves change places in Figure 4 after the first cycle at 400 rpm? The curve for the system with 0.025% CNC is higher compared to 0.05% CNC.
Response 4: Thank you for pointing this out. The curves in the first cycle represent the evolution of the mean chord length without CNC. As it is the first cycle, there are some differences due to the reproducibility of the stirring and dispersion. Because of that, the CNC were not added in this cycle, but it was added in the second cycle, after completing the dispersion of cemment reaching the same mean chord length in all the curves, as it can be seen in figure 4 time 1200. After adding the CNC both curves have similar mean chord lenthg until time 1550 when the effect of the higher dose of CNC is shown causing a decrease in the mean chord length wich indicates a higher dispersion effect. We agree that clarifying this point will improve the manuscript, therefore, a sentence was added in page 7, lines 261-264.
Comment 5: 396. We can add CNC to 0.05%.
Response 5: Thank you for pointing this out. We agree with this comment and we have added "CNC to" in the sentence (page 12 line 405).
Comment 6: Add a period at the end of the sentence.
Reponse 6: Thank you for this comment. We have added a point as suggested (page 16, line 554)
Comment 7: The conclusions and the list of references are compiled in accordance with the requirements of the journal.
The manuscript explains in a simple form the method of replacing SP with CNC, i.e. it suggests a transition from the use of synthetic materials to natural and constantly renewable ones. I think this is interesting, especially since the results obtained by the authors have a positive assessment. I think the manuscript deserves the right to be published.
Response 7: Thank you for this comment. We are happy you consider the article interesting.